# Verbal Encoding Deficits Impact Recognition Memory in Atypical “Non-Amnestic” Alzheimer’s Disease

**DOI:** 10.3390/brainsci12070843

**Published:** 2022-06-28

**Authors:** Deepti Putcha, Nicole Carvalho, Sheena Dev, Scott M. McGinnis, Bradford C. Dickerson, Bonnie Wong

**Affiliations:** 1Frontotemporal Disorders Unit, Massachusetts General Hospital and Harvard Medical School, Boston, MA 02115, USA; ncarvalho3@mgh.harvard.edu (N.C.); sdev25@gmail.com (S.D.); smmcginnis@bwh.harvard.edu (S.M.M.); brad.dickerson@mgh.harvard.edu (B.C.D.); bonnie.wong@mgh.harvard.edu (B.W.); 2Department of Psychiatry, Massachusetts General Hospital and Harvard Medical School, Boston, MA 02115, USA; 3Center for Brain Mind Medicine, Brigham and Women’s Hospital and Harvard Medical School, Boston, MA 02115, USA; 4Department of Neurology, Massachusetts General Hospital and Harvard Medical School, Boston, MA 02115, USA; 5Athinoula A. Martinos Center for Biomedical Imaging, Massachusetts General Hospital and Harvard Medical School, Boston, MA 02115, USA

**Keywords:** episodic memory, atypical AD, PCA, lvPPA, verbal learning

## Abstract

Memory encoding and retrieval deficits have been identified in atypical Alzheimer’s disease (AD), including posterior cortical atrophy (PCA) and logopenic variant primary progressive aphasia (lvPPA), despite these groups being referred to as “non-amnestic”. There is a critical need to better understand recognition memory in atypical AD. We investigated performance on the California Verbal Learning Test (CVLT-II-SF) in 23 amyloid-positive, tau-positive, and neurodegeneration-positive participants with atypical “non-amnestic” variants of AD (14 PCA, 9 lvPPA) and 14 amnestic AD participants. Recognition memory performance was poor across AD subgroups but trended toward worse in the amnestic group. Encoding was related to recognition memory in non-amnestic but not in amnestic AD. We also observed cortical atrophy in dissociable subregions of the distributed memory network related to encoding (left middle temporal and angular gyri, posterior cingulate and precuneus) compared to recognition memory (anterior medial temporal cortex). We conclude that recognition memory is not spared in all patients with atypical variants of AD traditionally thought to be “non-amnestic”. The non-amnestic AD patients with poor recognition memory were those who struggled to encode the material during the learning trials. In contrast, the amnestic AD group had poor recognition memory regardless of encoding ability.

## 1. Introduction

Alzheimer’s disease (AD) has a heterogeneous clinical presentation, driven by heterogeneity in the localization of neurodegeneration. AD can now be diagnosed using validated in vivo biomarkers which serve as proxies for AD-related neuropathic changes: amyloid, tau, and neurodegeneration (ATN) [1,2,3], allowing researchers to investigate the entire AD syndromic spectrum rather than rely on specific clinical criteria. Apart from the typical amnestic presentation of AD, characterized by episodic memory impairment, atypical clinical variants of AD include posterior cortical atrophy (PCA) and logopenic variant primary progressive aphasia (lvPPA). Diagnostic criteria for both PCA and lvPPA propose that episodic memory is relatively preserved at initial presentation but may develop as the disease progresses [4,5]. Though little is known about how memory impairment emerges in atypical AD, a growing body of evidence suggests that memory deficits may be present earlier than previously understood. PCA, commonly thought of as a “visual variant of AD” [6] and characterized primarily by progressive decline in higher-order visual processing and other posterior cortical functions [4,7], can also present with impairment in auditory verbal working memory [8] and verbal episodic memory [9,10,11]. Similarly, although the core diagnostic criteria for PPA specify that memory should be relatively preserved, lvPPA patients may also exhibit verbal [12] and non-verbal episodic memory deficits [13]. Episodic memory can be understood as three stages: encoding, storage, and retrieval. Encoding is the process of attending to, perceptually processing, and organizing information to be learned. Storage refers to the ability to retain newly learned information over time, and retrieval refers to the ability to recall this specific information in a goal directed fashion. Often, retrieval is tested as the ability to freely recall the new information, while recognition memory (often yes/no forced choice) tests the ability to discriminate old versus new information, tapping into storage abilities. As visual and auditory–verbal processing is critical to memory encoding, and thus subsequent storage and retrieval, modality-specific memory impairment may be a salient feature of these atypical variants of AD, potentially attributable to inefficient perceptual processing during encoding. Therefore, there is a critical need to better characterize these memory deficits and understand the mechanisms by which they emerge. The current study builds on the existing literature in atypical AD by focusing on how encoding impairment may impact recognition memory which has been previously thought to be spared early in these atypical “non-amnestic” variants of AD compared to amnestic variants of AD.

Research over the past several decades in typical amnestic AD has recently been supported by multimodal imaging studies in demonstrating that the medial temporal lobes, and particularly the hippocampus, are central in supporting episodic memory function [14,15,16]. We have a growing understanding that distributed cortical regions also contribute critically to encoding and retrieval processes, including the prefrontal, temporoparietal, and medial and lateral parietal regions, which are parts of the dorsal attention, frontoparietal, default mode, and language networks [17,18]. The parietal cortex, in particular, has consistently been linked to successful memory encoding and retrieval; three regions within the parietal lobe—the posterior cingulate cortex (PCC), precuneus, and posterior inferior parietal lobule (pIPL)—have been dubbed the “parietal memory network” [19]. In contrast, recognition memory, thought to call upon the dual processes of recollection and familiarity, has long been understood to be reliant on the hippocampus and extrahippocampal medial temporal lobe structures, specifically the perirhinal and lateral entorhinal cortices [18,20,21,22]. Some parietal cortical regions have also been identified as supporting recognition memory processes [23]. Specifically, the lateral temporoparietal cortex and the posterior cingulate cortex have been linked with recollection, while the superior parietal cortex and the precuneus support familiarity [24]. Further, regions within the intraparietal sulcus and inferior parietal lobule, as well as midline structures that extend from the retrosplenial cortex and posterior cingulate cortex to the precuneus, show increased activity to recognized old items (hits) and mistakenly recognized new items (false alarms), compared with correctly rejected new items and forgotten old items (misses); this phenomenon has been described as the “old/new” effect [23].

Patients with AD pathology, irrespective of clinical presentation, demonstrate reduced cortical thickness bilaterally in the posterior cingulate cortex and precuneus [25,26], regions comprising the so-called parietal memory network [19]. Because these medial and lateral parietal (as well as lateral temporal) cortical regions are heavily impacted in atypical phenotypes of AD, we expect to observe diminished memory storage in addition to encoding and retrieval impairment. Indeed, a recent investigation of verbal list learning performance in PCA revealed impaired recognition memory comparable to that seen in typical amnestic AD [9]. Another recent study from the same group suggested reduced cue sensitivity in PCA compared to healthy control participants [11], pointing again to memory storage loss. In contrast, recognition memory in lvPPA patients has been previously reported to be normal [12,27].

The focus of the present study is to investigate verbal recognition memory performance across the amyloid- and tau-positive spectrum of AD. Specifically, we seek to determine how the atypical “non-amnestic” phenotypes of AD—PCA and lvPPA—compare to amnestic AD on verbal list learning, retrieval, and recognition memory performance. Another goal is to evaluate how difficulties during the encoding phase of a list learning test (related in part to impaired perceptual processing or working memory) predict recognition memory performance in both AD groups. Lastly, we investigate the relationships between memory performance and patterns of cortical atrophy in the entire sample. We hypothesize that the “non-amnestic” AD group will have stronger recognition memory performance compared to amnestic AD, though performance is likely to be impaired across AD syndromes. We also hypothesize that stronger recognition discriminability will be a function of stronger encoding performance across groups, though we will likely observe a wider range of recognition memory performance in non-amnestic AD compared to amnestic AD. Based on our understanding of network dysfunction underlying memory impairment in amnestic AD, we predict that cortical atrophy in widespread temporoparietal and frontal cortical regions will relate to encoding performance, while atrophy in predominantly medial temporal lobe regions will relate to recognition discriminability.

## 2. Materials and Methods

### 2.1. Participant Characteristics

Data for this study were obtained from thirty-seven symptomatic A + T + N + individuals who were recruited from the Massachusetts General Hospital (MGH) Frontotemporal Disorders Unit, including the PPA program [28] and PCA program [10,29]. See Table 1 for full demographic data. All patients received a standard clinical evaluation comprising a comprehensive neurological and psychiatric history and exam, structured informant interviews following the Clinical Dementia Rating (CDR) protocol, a neuropsychological battery including the National Alzheimer’s Coordinating Center (NACC) Uniform Data Set (UDS) version 3.0 battery, Frontotemporal Lobar Degeneration (FTLD) module, and an additional MGH FTD Unit testing battery focusing on memory and visuospatial cognition. For each patient, clinical diagnostic formulation was performed through consensus conference within our multidisciplinary team, with each patient being classified based on all clinical information as having mild cognitive impairment or dementia (cognitive functional status), and then each patient’s cognitive–behavioral syndrome being diagnosed according to standard diagnostic criteria [30]. Fourteen patients met diagnostic criteria for PCA [4,7,31], and nine patients met criteria for lvPPA [5]. All of these patients exhibited relative preservation of memory on the structured examination of the patient and interview with the informant (including the CDR protocol autobiographical episodic memory interview). The remaining 14 patients were classified as amnestic AD (amnestic AD), as they presented with a predominant amnestic syndrome (informant-corroborated symptoms of memory loss in daily life and impaired memory on examination and interview, including the CDR memory interview) and a secondary dysexecutive syndrome (reported executive dysfunction and impaired scores on tests of working memory, set-shifting, and response inhibition).

Participants also underwent neuroimaging sessions which included a high-resolution 3 Tesla MRI, 18F-AV-1451 positron emission tomography (tau PET), and 11C Pittsburgh Compound B (PiB) positron emission tomography (amyloid PET) imaging. We only included patients in this study who had a positive amyloid PET scan (Aß+), as assessed by visual read according to previously published procedures [32], and met biomarker criteria for MCI [33] or dementia [2] highly likely due to AD. Each of these individuals also had tau-positive and neurodegeneration-positive imaging by visual read. Individuals were excluded from this cohort if they had a primary psychiatric or other neurologic disorder. This work was carried out in accordance with The Code of Ethics of the World Medical Association (Declaration of Helsinki) for experiments involving humans. All participants and their informants/caregivers provided informed consent in accordance with the protocol approved by the Mass General Brigham Human Research Committee Institutional Review Board in Boston, Massachusetts.

### 2.2. Memory Testing and the Neuropsychological Battery

Within two months from the PET and MR scans, all Aß+ participants underwent a neuropsychological task battery assessing a range of cognitive skills. The cognitive function of interest, verbal episodic memory, was assessed using the California Verbal Learning Test II-Short Form (CVLT-II-SF) [34]. The CVLT-II-SF is a nine-item list learning test composed of words belonging to three semantic categories: clothing, fruit, and tools. The word list is repeated over four encoding trials. After a brief 30 s delay, a short-delay free recall (SDFR) condition is administered. After a longer 10 min delay, the long-delay free recall (LDFR) and long-delay cued recall (LDCR) conditions are administered followed immediately by recognition testing which is in a yes/no recognition format and includes 9 target (“hits”) and 18 foils (9 semantically related and 9 semantically unrelated words). Raw scores (Table 2) and demographically-adjusted z-scores (Appendix A) were calculated for Trial 1, total encoding (sum of trials 1–4), SDFR, LDFR, LDCR, recognition discriminability (d’), and response bias (C).

Global cognitive function was assessed using the Montreal Cognitive Assessment (MoCA), and comprehensive neuropsychological performance was evaluated with a combination of the NACC UDS version 3 protocol and the supplemental MGH FTD unit battery. We assessed auditory attention and working memory using the NACC UDS3 Number Span Forward and Backward subtests. Visuomotor sequencing was evaluated with Trail Making Test Part A and visuomotor set-shifting was evaluated with Trail Making Test Part B. Confrontation naming was evaluated in the auditory (Auditory Naming Test; ANT) and visual (Multilingual Naming Test; MINT) modalities. Verbal fluency was evaluated with letter fluency trials (FAS) and a category fluency trial (Animals). Sentence repetition and reading were evaluated as part of the NACC UDS version 3 FTLD module. Verbal memory was also evaluated with the Craft Story immediate and delayed recall conditions and the Benson Figure delayed recall. Visual attention was evaluated with the Visual Object and Space Perception (VOSP) Number-Location Test and visual construction was tested with a clock drawing test as part of the MoCA and the Benson Figure Copy.

Performance differences on the CVLT-II-SF between the Aß+ groups were investigated using independent samples t-tests. Given that the PCA and lvPPA groups performed comparably on total encoding and recognition memory (d’ and C), our outcome measures of interest, these groups were combined into one atypical “non-amnestic” subgroup (Appendix A) to compare against the amnestic subgroup. The association between total encoding and recognition discriminability (d’) was evaluated using Pearson’s bivariate correlation analyses, as was the relationship of each of these measures with the number span subtests. For these analyses, as well as group comparisons on demographic factors, alpha was set at 0.05. Primary hypothesis-driven analyses were conducted on these two primary outcome measures without correction for multiple comparisons. Statistical analyses were conducted in IBM SPSS Version 24.0 (Armonk, NY, USA).

### 2.3. Neuroimaging Data Acquisition and Analysis

All participants underwent 18F-AV-1451 (tau) and 11C- Pittsburgh Compound B (amyloid) PET scans. The 18F-AV-1451 radiotracer was prepared at MGH with a radiochemical yield of 14% ± 3% and specific activity of 216 ± 60 GBq/μmoL (5837 ± 1621 mCi/μmoL) at the end of synthesis (60 min) and validated for human use [35]. Scans were acquired from 80 to 100 min after a 10.0 ± 1.0 mCi bolus injection in 4 × 5 min frames. The 11C-PiB PET radiotracer was acquired with an 8.5 to 15 mCi bolus injection followed immediately by a 60 min dynamic acquisition in 69 frames (12 × 15 s, 57 × 60 s). All PET data were acquired using a Siemens/CTI (Knoxville, TN, USA) ECAT HR+ scanner (3D mode; 63 image planes; 15.2 cm axial field of view; 5.6 mm transaxial resolution; 2.4 mm slice interval). Data were reconstructed and attenuation corrected; each frame was evaluated to verify adequate count statistics; interframe head motion was corrected prior to further processing.

All participants also underwent a high-resolution MRI scan (Siemens TIM Trio 3.0 Tesla) with an average time delay of 21 ± 23 days from PET scans, and included acquisition of T1-weighted multi-echo magnetization prepared rapid acquisition gradient echo (MPRAGE) structural images. The MRI analysis methods employed here have been previously described in detail [36,37], including cortical thickness processing and spherical registration to align subjects’ cortical surfaces. Visual inspection confirmed accurate registration between anatomical and PET volumes. To evaluate the anatomy of PET binding, each individual’s PET dataset was rigidly co-registered to the subject’s MPRAGE data. Similar to a previous report, 18F-AV-1451 specific binding was expressed as the standardized uptake value ratio (SUVR), using the whole cerebellar grey matter as a reference [38]. 11C-PiB PET data were expressed as the distribution volume ratio (DVR) with the cerebellar grey matter as a reference [39], where regional time–activity curves were used to compute regional DVRs for each ROI using the Logan graphical method applied to data from 40 to 60 min after injection. PET data were partial-volume-corrected and were processed using geometric transform matrix as implemented in FreeSurfer stable release version 6.0.

To determine if demographically-adjusted memory task performance (z-scores of total encoding and recognition discriminability) was related to cortical atrophy in hypothesized regions, statistical surface maps were generated by computing a general linear model (GLM) for the effects of the task performance on cortical thickness at each vertex point on the cortical surface using methods we have previously published [40,41]. We used age-, education-, and sex-adjusted z-scores for memory task performance and thus did not control for these demographic factors again in our cortical thickness GLM analysis. Follow-up analysis ensured that cortical thickness was not related to any of these demographic factors. GLM analyses were implemented using the mri_glmfit utility within FreeSurfer version 6.

## 3. Results

### 3.1. Clinical and Cognitive Characteristics

Thirty-seven Aß+ patients (14 amnestic, 14 PCA, and 9 lvPPA) were included in this study (Table 1). The amnestic AD group was younger than the PCA (t = 3.3, *p* = 0.003) and lvPPA (t = 3.4, *p* = 0.003) groups, and the PCA and lvPPA groups were comparable in age (t = 0.25, *p* = 0.8). There were no statistically significant differences between groups in education, sex, or handedness. The majority of these Aß+ patients were given a global CDR of 0.5, consistent with mild cognitive impairment.

All participants completed a neuropsychological testing battery (Table 3). The mean MoCA score by group was 14.9 in amnestic AD, 17.9 in PCA, and 15.5 in lvPPA (ANOVA: F = 0.95, *p* = 0.40). MoCA scores in each group were statistically comparable to each other indicating equivalent severity of global cognitive deficits: amnestic vs. PCA, t = 1.3, *p* = 0.21; amnestic vs. lvPPA, t = 0.25, *p* = 0.80; PCA vs. lvPPA, t = 0.95, *p* = 0.35. Performances on a more comprehensive battery of neuropsychological assessments were consistent with each syndromic group’s predominant clinical presentation. The amnestic group performed more poorly compared to PCA on number span forward (t = 2.9, *p* = 0.008), auditory confrontation naming (t = 2.1, *p* = 0.04), letter fluency (t = 4.6, *p* = 0.0002), immediate story memory (t = 2.4, *p* = 0.02), and delayed story memory (t = 3, *p* = 0.006). In contrast, the PCA group performed worse than the amnestic group on Trails A (t = 3.8, *p* = 0.002), visual confrontation naming (t = 2.3, *p* = 0.03), and figure copy (t = 3.6, *p* = 0.002). The lvPPA group performed worse than the amnestic group on auditory confrontation naming (t = 2.5, *p* = 0.02) and visual confrontation naming (t = 2.9, *p* = 0.008), but better than the amnestic group on story memory recall (t = 2.5, *p* = 0.02). Finally, the PCA group performed better than the lvPPA group on number span forward (t = 3.6, *p* = 0.002), auditory confrontation naming (t = 4.4, *p* = 0.0002), sentence repetition (t = 2.5, *p* = 0.04), and immediate story memory (t = 2.6, *p* = 0.02), but more poorly than the lvPPA group on Trails A (t = 7.0, *p* = 0.000009) and figure copy (t = 5.6, *p* = 0.00006).

### 3.2. CVLT-II-SF Performance

Scores on the CVLT-II-SF verbal list learning test are reported in Table 3. Performance on Trial 1 was impaired in the lvPPA group compared to PCA group (t = 2.6, *p*= 0.02; Appendix A), with the amnestic group showing a trend toward impairment compared to PCA (t = 2.0, *p* = 0.06). On total encoding, the amnestic group performed poorly compared to the PCA group (t = 2.6, *p* = 0.01), and comparable to the lvPPA group. The amnestic group was impaired compared to the lvPPA group on long-delay free recall (t = 2.7, *p* = 0.01). There were no differences between the PCA and lvPPA group on our two metrics of interest in this study, total encoding (t= 1.6, *p*= 0.13) and recognition discriminability (t = 1.5, *p* = 0.9); see Appendix A. Thus, we combined the groups together into a “non-amnestic” group for all subsequent analyses. There were no other statistically significant between-group differences observed on the remainder of the CVLT-II-SF measures. Comparing the non-amnestic and amnestic groups, we observed better performance in the non-amnestic group only on LDFR (t = 2.1, *p* = 0. 04) and LDCR (t = 2.1, *p* = 0.05) with a trend-level difference on total encoding (t = 1.9, *p* = 0.06) and recognition discriminability (t = 1.8, *p* = 0.08; Figure 1A).

### 3.3. Recognition Memory Is Impaired across Atypical Aß + AD Syndromes

Recognition memory performance was evaluated by examining recognition discriminability (d’) and response bias (C). In the context of a perfect d’ score being 3.5 (i.e., 9/9 hits with no false positive errors), performance in both groups was poor (Figure 1A; amnestic group mean = 1.3, non-amnestic group mean = 1.9). Given that d’ is a measure encompassing both correctly identified target words (hits) as well as incorrectly endorsed foil words (false positives), we further investigated group differences on these specific metrics. We found that the groups performed similarly on hits (t = 0.37, *p* = 0.7) but the non-amnestic group endorsed fewer false positive items compared to the amnestic group (t = 0.20, *p* = 0.05; Appendix A). The non-amnestic group also demonstrated a neutral response bias while the amnestic group showed a substantially liberal response bias (the group comparison revealed a trend toward a difference, t= 1.6, *p* = 0.1; Figure 1B). Follow-up analyses in the non-amnestic group revealed only a trend level difference in percentage of false positive errors compared to false negative errors (t = 1.8, *p* = 0.08; Appendix A). In contrast, the amnestic group demonstrated a significantly higher percentage of false positive compared to false negative responses (t = 3.3, *p* = 0.006).

### 3.4. Total Encoding Is Related to d’ in Non-Amnestic Atypical AD

Next, we sought to determine whether the ability to encode information efficiently was related to d’. We found that the non-amnestic group demonstrated a positive learning curve across the four repeated trials of the CVLT-SF but the amnestic group did not (Figure 2A). Both the amnestic and the non-amnestic groups demonstrated poor Trial 1 encoding (t = 0.69, *p* = 0.5). However, the non-amnestic group benefitted from repetition of the word list across trials while the amnestic group did not. By Trial 4, the non-amnestic group performed better than the amnestic group (t = 2.6, *p* = 0.01). When we examined the association between total encoding performance (sum of Trials 1 through 4) and d’ (Figure 2B), we found a strong positive relationship between total encoding and recognition d’ in the non-amnestic group (r = 0.72, *p* = 0.0002) but no association between these two phases of the list learning task in the amnestic group (r = 0.12, *p* = 0.7).

Follow-up analyses investigating the role of auditory–verbal working memory in the different stages of verbal memory performance were conducted by correlating performance on the number span backward test to total encoding and d’ performance (Appendix A). We found that number span backward performance was related to total encoding performance in both the amnestic group and the non-amnestic group, such that stronger working memory was related to stronger encoding. In contrast, we did not find a relationship between number span backward performance and d’ in either AD subgroup.

### 3.5. Retention Is Related to d’ in Non-Amnestic AD

Next, we sought to determine if the amount of information retained was related to d’ in an effort to provide converging evidence for our hypothesis that amnestic AD shows evidence of memory storage loss. We defined percent retention as the number of words recalled at long-delay free recall divided by the number of words recalled at Trial 4 of the encoding phase. Comparing the amnestic AD to non-amnestic AD, we found that the groups did not differ statistically in the amount of information retained after a long delay (amnestic mean = 41%, non-amnestic mean = 57%; t = 1.08, *p* = 0.29). When we directly investigated the association between retention and d’ (Appendix A), we found a strong positive relationship in non-amnestic AD (r = 0.61, *p* = 0.005), but no association between these two metrics in amnestic AD (r = 0.26, *p* = 0.40).

### 3.6. Total Encoding and d’ Are Related to Atrophy in Dissociable Cortical Regions

We tested our a priori hypotheses regarding the neuroanatomical correlates of memory encoding and recognition discriminability by conducting two separate whole-cortex GLMs predicting performance on total encoding (sum of Trials 1–4) and recognition discriminability (d’), respectively (Figure 3). We included all A + T + N + participants (n = 37) in these analyses to capitalize on the heterogeneity in clinical presentation and neurodegeneration across the groups. We found associations between total encoding and cortical atrophy in several regions known to support memory encoding based on previous work in typical older-onset amnestic AD [17,18], including predominantly left-lateralized middle temporal gyrus, pIPL (angular gyrus), posterior cingulate cortex, and precuneus (Figure 3A). Medial temporal lobe atrophy did not relate to total encoding performance. In contrast, we observed a circumscribed association between d’ and cortical atrophy primarily in bilateral anterior medial temporal cortices (Figure 3B).

## 4. Discussion

Episodic memory deficits are commonly reported across the heterogeneous clinical syndromes of AD. While individuals with an amnestic syndrome typically have impairments in both memory storage and retrieval (and sometimes encoding), the memory profiles of the atypical “non-amnestic” presentations of AD (PCA, lvPPA) and their anatomical underpinnings have been less clearly understood. This may be, in part, because the diagnostic criteria of PCA and lvPPA specifically require a relative preservation of episodic memory. However, recent work in atypical variants of AD has demonstrated that memory encoding and/or delayed recall is often impaired in PCA [10,42] and lvPPA [12,13]. Because memory storage requires that material be encoded, we hypothesized here that delayed recognition discriminability may also be impaired in PCA and lvPPA. That is, we predicted that impaired recognition discriminability in atypical AD would not represent memory storage loss but rather ineffective encoding. Last, we predicted that these memory processes would be related to dissociable patterns of cortical atrophy consistent with what has been described in the literature on amnestic AD syndromes.

We first report evidence that recognition memory is impaired in this group of patients with non-amnestic AD syndromes, in addition to patients with amnestic AD, relatively early in the course of illness. The characteristics of impaired recognition discriminability in the amnestic group were typical of amnestic patients, with poor discriminability and a liberal response bias (tendency to endorse more words as familiar than not), consistent with prior reports indicating a liberal response bias in AD [43,44]. The non-amnestic group was comparatively less impaired on discriminability and exhibited a neutral response bias. Impaired recognition memory has previously been reported in PCA at levels comparable to amnestic AD [9,42], and one report even demonstrated a liberal response bias on the Rey Auditory Verbal Learning Test (Ahmed et al., 2018), which is a similar list learning test to the CVLT-II. In addition, reduced cue sensitivity has been identified in PCA compared to healthy older control participants [11]. When examining the PCA and lvPPA groups separately, we found that these groups were impaired at a level comparable to each other on recognition memory performance, suggesting that this is a common feature across these atypical AD variants. This observation in lvPPA contrasts with prior work reporting normal verbal recognition memory performance in lvPPA [12], though global cognitive impairment of the participants in that study was milder than in our study. Taken together, we report that recognition memory deficits on a list learning task may emerge in some patients with atypical “non-amnestic” variants of AD at clinical stages of MCI or mild dementia. The yes/no recognition discriminability task employed in this study is less vulnerable to the executive retrieval demands that typically confound measures of delayed recall and retention traditionally used to measure storage and retrieval ability in these populations. However, impaired recognition memory performance does not necessarily reflect impaired memory storage and could reflect inefficiencies at other stages of memory.

To better understand the cognitive mechanisms underlying impaired recognition memory in atypical AD syndromes, we investigated the association between memory encoding and recognition discriminability. We hypothesized that poor encoding ability would be related to poor recognition discriminability after a delay period across groups, supporting the idea that one cannot store what one does not encode. We observed a wide range of performance on both encoding and recognition discriminability, though the average encoding and recognition discriminability scores were impaired across AD syndromes. The non-amnestic group differed from the amnestic group in that they benefited from repetition of encoding trials while the amnestic group demonstrated a largely flat learning curve. We found that total encoding and recognition discriminability performance were positively associated with each other only in the non-amnestic AD group, such that worse recognition memory performance was likely driven by poorer encoding. Further, we found that verbal encoding success was directly related to auditory-verbal working memory across AD groups, while recognition discriminability was not, which is consistent with our previous work describing the association between auditory–verbal working memory and list learning in PCA [10]. Together, these results suggest that those non-amnestic individuals who were able to encode the word list successfully after four learning trials were able to store this information over time, while those who struggled to encode the word list could not discriminate target words from foils on a relatively easy yes/no forced-choice recognition test. Given that recognition discriminability measures do not rely on verbal retrieval ability, which is known to be impaired in PCA and lvPPA [10,12,17,42], we conclude that ineffective encoding leads to an impaired ability to subsequently discriminate novel from learned information in these atypical AD syndromes. We did not find an association between encoding and recognition discriminability in the amnestic group, which may have been due, in part, to “pure” storage loss in this group (i.e., no matter how well these people encode information, they do not store it for later recall or recognition). The amnestic group demonstrated a smaller range of discriminability performance, indicating poor storage in this group irrespective of the level of encoding success.

We then examined the neuroanatomical correlates of each of these memory stages across the AD syndromic spectrum. Consistent with previous reports from our group (Putcha et al., 2019; Wolk et al., 2011) on typical older-onset amnestic AD, we observed that total encoding was associated with cortical atrophy in predominantly left-hemisphere temporoparietal cortex: middle temporal gyrus, pIPL (angular gyrus), posterior cingulate cortex, and precuneus. These regions, which represent parts of the frontoparietal control network, default mode network, and language network, have been well established as supporting the multidomain cognitive factors associated with strategic encoding and retrieval [45,46]. We did not observe any relationship between medial temporal lobe atrophy and total encoding performance. In our prior work, initial (Trial 1) encoding performance in typical late-onset amnestic AD showed a similar pattern, but we previously found that total encoding of a word list is associated with medial temporal lobe atrophy (e.g., Putcha et al., 2019; Wolk et al., 2011). This discrepancy may be due to the younger age of many of the patients in this sample compared to the range of typical older-onset AD patients who may have greater medial temporal lobe atrophy and memory performance as a function of age alone. In direct contrast, we observed a very circumscribed association between d’ and cortical atrophy primarily in bilateral anterior medial temporal cortices, consistent with decades-long knowledge underscoring the critical role these areas play in memory storage.

Our study has several strengths, including the focus on atypical variants of AD which are typically understudied populations. Further, we selected only biomarker-positive (A + T + N +) individuals, increasing our confidence that these individuals with atypical phenotypic presentations were indeed those with underlying AD pathology. Furthermore, we studied a verbal list learning task that is very commonly administered in the clinic (CVLT-II-SF), with the goal of directly informing clinical decision-making and increasing the translational value of this work. Despite these strengths, some limitations to this study should be addressed. First, we were not able to study item-level data to determine the exact correspondence between the specific words encoded reliably across learning trials and retention or recognition of those same words. Such an approach could add valuable information to further substantiate our conclusions regarding the relationship between encoding and recognition discriminability and will be kept in mind in the design of future studies. Additionally, we are constrained by the relatively small sample size inherent to studies of these atypical presentations of AD (PCA, lvPPA) given the relatively rare incidence in the population. However, given the robust nature of our observations, we are confident in our interpretation of results and plan to build on this work as our dataset grows.

In summary, results from the present study demonstrate verbal list-learning memory impairment in some patients with atypical syndromes of AD, calling into question the categorization of these patients as “non-amnestic”. While these patients do not tend to have pure storage loss (i.e., loss of learned information over time), they could struggle with a verbal list-learning task for reasons related to problems with encoding which relies on working memory, language, and executive function skills. That is, the atypical AD patients who demonstrate difficulty on recognition memory are those who struggled to encode the information in the first place. These results suggest that verbal memory test performance needs to be contextualized in relation to other cognitive domains (e.g., working memory), which will ultimately improve our characterization of all AD clinical syndromes. Furthermore, these results have important clinical implications informing training of strategic encoding skills to benefit day-to-day memory, optimize functional independence, and improve quality of life.

## Figures and Tables

**Figure 1 brainsci-12-00843-f001:**
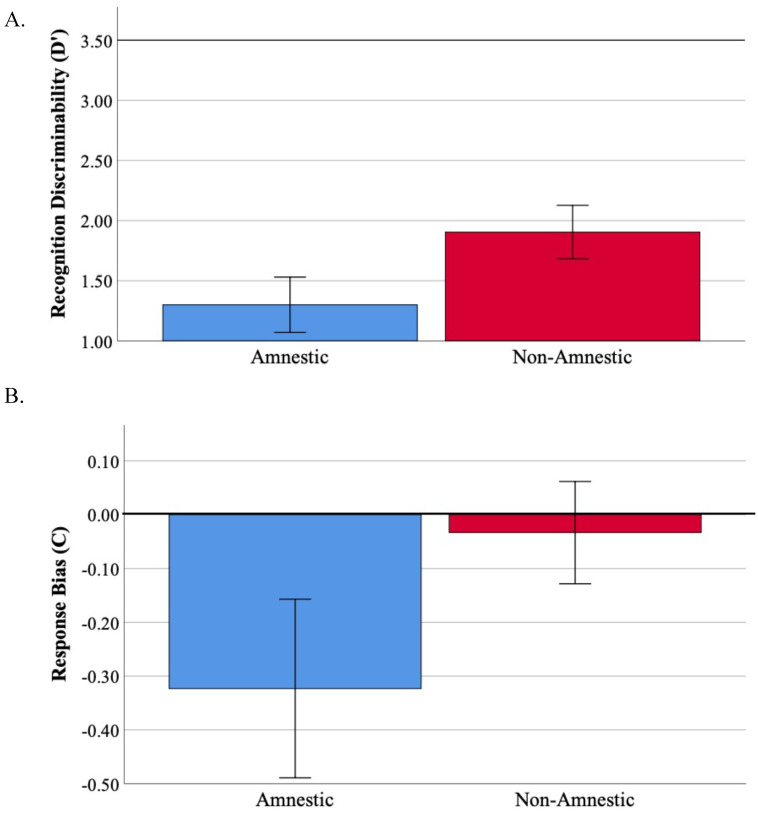
Recognition memory performance in AD. (**A**) Recognition discriminability (d’) is poorer in the amnestic compared to the non-amnestic group at a trend level (t = −1.8, *p* = 0.08, Cohen’s d = 0.64), though d’ is poor in both groups (a perfect d’ score is 3.5). (**B**) Response bias (C) is substantially liberal in the amnestic compared to non-amnestic group at a trend level (t = 1.6, *p* = 0.10, Cohen’s d = 0.5).

**Figure 2 brainsci-12-00843-f002:**
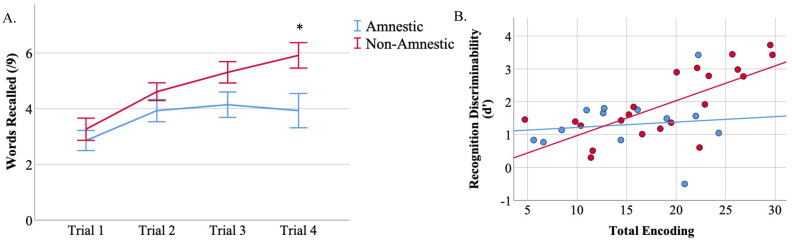
Total encoding is related to d’ in non-amnestic AD. (**A**) The non-amnestic group demonstrated poor Trial 1 learning comparable to the amnestic group. However, the non-amnestic group benefitted from repetition of the word list while the amnestic group demonstrated a flat learning curve. * *p* < 0.05. (**B**) Total encoding performance (sum of Trials 1 through 4) is related to recognition discriminability (d’) in the non-amnestic group (r = 0.72, *p* = 0.0002) but not the amnestic group (*p* = 0.7).

**Figure 3 brainsci-12-00843-f003:**
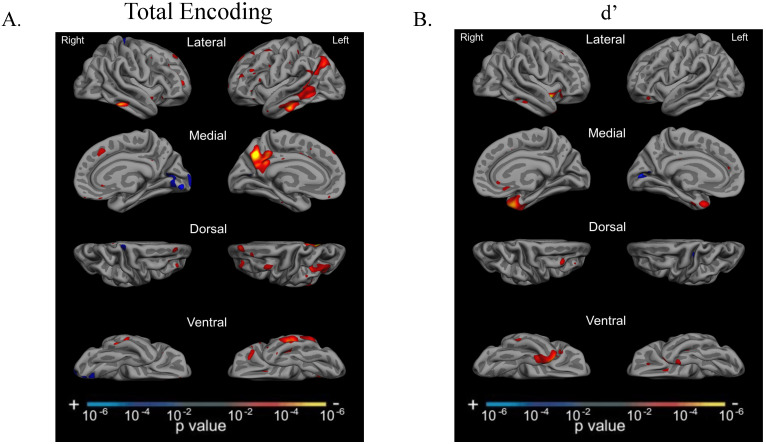
Total encoding and recognition discriminability (d’) are related to dissociable regions of atrophy. (**A**) Total learning (sum of Trials 1–4) was related to cortical atrophy in the left-hemisphere lateral middle and inferior temporal gyrus, as well as angular gyrus and posterior cingulate cortex/precuneus regions in the entire sample of AD participants at a threshold of *p* < 0.01. (**B**) Recognition discriminability was related to cortical atrophy in the left-hemisphere anterior medial temporal lobe across AD participants at a threshold of *p* < 0.01.

**Table 1 brainsci-12-00843-t001:** Demographic characteristics. Mean (SD) presented for each continuous demographic factor.

Demographic	All (n = 37)	Amnestic (n = 14)	PCA (n = 14)	lvPPA (n = 9)
Age (years)	67.1 (8.7)	61.2 (5.9)	70.3 (8.4) *	71.2 (8.3) *
Sex Ratio (M:F)	16M/21F	7M/7F	5M/9F	4M/5F
Education (years)	16.4 (2.6)	16.9 (2.2)	16.8 (2.1)	14.9 (3.5)
Handedness (R:L)	34R/3L	12R/2L	14R/0L	8R/1L
CDR Global	CDR 0 (N = 2) CDR 0.5 (N = 21) CDR 1 (N = 14)	CDR 0.5 (N = 8) CDR 1 (N = 6)	CDR 0.5 (N = 7) CDR 1 (N = 7)	CDR 0 (N = 2) CDR 0.5 (N = 6) CDR 1 (N = 1)

M = male, F = female, R = right-handed, L = left-handed; CDR= Clinical Dementia Rating. * Indicates a statistical difference between this group and the amnestic AD group at *p* < 0.05. There were no statistical differences observed between PCA and lvPPA groups at *p* < 0.05.

**Table 2 brainsci-12-00843-t002:** CVLT-II-SF Performance. Mean (SD) of raw scores presented for each measure.

CVLT-II-SF	All (n = 37)	Amnestic (n = 14)	PCA (n = 14)	lvPPA (n = 9)
Trial 1	3.1 (1.7)	2.9 (1.3) *	4.0 (1.7) ^	2.1 (1.7) *^
Total Encoding (Sum Trials 1–4)	17.4 (6.6)	14.9 (5.9) *	20.7 (5.6) †	16.3 (7.6) *
SDFR	3.7 (2.2)	2.9 (1.9) *	4.4 (2.5)	4.0 (2.0) *
LDFR	2.8 (2.6) †	1.7 (1.9) *	3.1 (2.9)	4.1 (2.4) †
LDCR	3.2 (25)	2.1 (1.9) *	3.8 (2.7) *	3.9 (2.8) *
Recognition Discriminability (d’)	1.7 (1.0)	1.3 (0.8)	1.9 (1.1)	1.9 (1.0)
Response Bias (C)	−0.1 (0.5)	−0.3 (0.6)	−0.04 (0.3)	−0.02 (0.6)

SDFR = Short Delay Free Recall. LDFR = Long Delay Free Recall. LDCR = Long Delay Cued Recall. d’ = Recognition Discriminability. C = Response Bias. * indicates impairment at the level of 1.5 SD below the mean. † indicates differences between this group and the Amnestic Group at *p* < 0.05. ^ indicates differences between PCA and lvPPA groups at *p* < 0.05.

**Table 3 brainsci-12-00843-t003:** **Neuropsychological Profiles.** Mean (SD) presented for each test score.

Test	All (n = 37)	Amnestic (n = 14)	PCA (n = 14)	lvPPA (n = 9)
MoCA (out of 30)	16.2 (6.0)	14.9 (6.1)	17.9 (6.2)	15.5 (5.2)
**Attention/Executive Functions**				
Longest Digit Span Forward	5.6 (1.7)	5.2 (1.1)	6.7 (1.6) *^	4.3 (1.5) ^
Longest Digit Span Backward	3.3 (1.4)	2.9 (1.4)	3.8 (1.4)	3.2 (1.0)
Trails A ^a^	76.5 (43.9)	65.7 (36.4)	127 (24.9) *^	44.4 (20.5) ^
Trails B ^b^	200.1 (62)	237 (58.3)	164.5 (10.6)	190.3 (75)
**Language**				
Auditory Naming Test (/50)	42.1 (8.4)	42.6 (6.7)	46.9 (3.5) *^	33.8 (10.4) *^
Auditory Naming Test with PC	46.1 (5.6)	45.9 (5.3)	48.9 (1.8)	42 (7.6)
MINT (/31)	22.8 (7.8)	26.9 (4.5)	20.1 (9.7) *	20.1 (6.5) *
MINT with SC	23.3 (7.9)	27.1 (4.4)	24 (7.5)	20.2 (6.6) *
MINT with PC	26.6 (5.4)	28.5 (3.5)	26.2 (6.8)	24.2 (4.6) *
Letter Fluency (FAS)	28.1 (17.8)	19.3 (9.8)	44.5 (15.7) *	23.7 (22.3)
Category Fluency (Animals)	11.4 (5.7)	9.7 (5.4)	14.6 (6.1)	9.3 (3.1)
Sentence Repetition (/5)	3.7 (1.5)	3.7 (1.4)	4.8 (0.5) ^	2.8 (3.1) ^
Sentence Reading (/5)	4.5 (0.67)	4.5 (0.8)	4.3 (0.6)	4.5 (0.5)
**Memory**				
Craft Story Immediate (/44)	9.5 (5.3)	7.8 (4.2)	12.8 (5.9) *^	6.9 (3.0) ^
Craft Story Delayed Recall (/44)	6.2 (5.9)	2.7 (3.1)	9.5 (7.2) *	6.4 (3.7) *
Benson Figure Recall (/17)	5.5 (4.8)	3.2 (3.9)	3.8 (5.3)	9.0 (3.7) *
**Visuospatial**				
Clock Drawing (/3)	1.6 (0.7)	1.6 (0.9)	1.5 (0.7)	1.7 (0.5)
Benson Figure Copy (/17)	12.1 (5.6)	13.7 (4.6)	5.4 (5.1) *^	15.3 (1.5) ^
VOSP Number Location (/10)	5.8 (2.9)	6.1 (2.9)	4 (2.3) ^	7.8 (2.5) ^

MOCA = Montreal Cognitive Assessment. MINT= Multilingual Naming Test. SC = Semantic Cue. PC = Phonemic Cue. ^a^ Only 24/37 people were able to complete Trails A. ^b^ Only 9/37 people were able to complete Trails B. * indicates statistical differences between this group and the Amnestic group at *p* < 0.05. ^ indicates statistical differences between PCA and lvPPA groups at *p* < 0.05.

## Data Availability

The data presented in this study are available on request from the corresponding author.

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
