# Peer review of "Verbal Encoding Deficits Impact Recognition Memory in Atypical “Non-Amnestic” Alzheimer’s Disease"

_brainsci, 2022, doi:10.3390/brainsci12070843_

Round 1

Reviewer 1 Report

In this article, Putcha et al., examined performance on the CVLT-II-SF in 23 amyloid+, tau+ and neurodegeneration+ participants with atypical non-amnestic AD and 14 amnestic AD participants. They showed diminished recognition memory performance in all groups, with the amnestic AD group performing slightly worse than the others. Encoding was related to recognition memory in non-amnestic AD but not in amnestic AD, suggesting poor storage abilities, independent of encoding, in amnestic AD but not in non-amnestic AD. They also looked at associations between cortical atrophy, using cortical thickness, and total encoding/recognition discriminability. These two measures were related to different atrophy patterns. They concluded that recognition memory is not spared in all patients with atypical AD thought to be “non-amnestic”.

First, I would like to congratulate the authors on their work. The research question they aimed to answer with this project is important and original. The methods used are robust and the findings are very interesting. Please find below some conceptual questions/suggestions to the authors.

Abstract:

·      Typo: 23 amyloid-, tau- should be 23 amyloid+, tau+

Introduction:

·      Before providing an in-depth description of memory deficits and their anatomical substrates in AD variants, I suggest briefly describing the concepts of encoding, storage, retrieval and recognition. This would be helpful to understand how each component can be differently affected in AD phenotypes.

Methods:

·      This section is well-described. However, I’d suggest trying to reduce the amount of text and place some information in supplementary, especially in the participant characteristics section.

Results:

·      Figure 2 is very small. It would be helpful to increase the size of the images.

·      The color bar on Figure 3 is also very small. It would be helpful to increase its size to see the range of p values.

·      The analyses conducted in the present study, as well as the choice of variables used or computed, are robust and well-justified. However, it would have been very interesting to look at item-level data to better understand the relationship between encoding and recognition. For instance, it would have allowed the authors to verify if words that are encoded 1-2-3 or 4 times were recognized or not, for all participants. One could hypothesize that individual words encoded 4 times have more chance to be recognized in non-amnestic AD, but not in amnestic AD. Looking at item-level data would have also allowed to differentiate between recognition memory deficits due to lack of encoding versus pure storage deficit due to loss of information over time (i.e. word is encoded but is lost over time). This should either be included as a new analysis or mentioned in the limitations section.

Discussion:

·      The discussion is well-organized, and the authors did a great job at discussing their results in the context of previous findings.

·      Conclusion: I think it would be helpful to rephrase the conclusion so that readers understand the difference between recognition memory deficits due to loss of information over time versus due to poor encoding (the two components being the core memory deficits in amnestic AD). I agree that using the term “non-amnestic” can be confusing because atypical AD patients can show impairments in recognition, but they don’t show loss of information for words encoded multiple times, which is what differentiates these patients from amnestic AD. My understanding from the findings is that non-amnestic AD patients can have difficulty learning new information, but once it is recalled it is not lost, compared to amnestic AD patients, who rather lose the information whether it is recalled or not. Rephrasing the conclusion to explain this important distinction would be helpful.

Author Response

Please see attached cover letter with all reviewer comments addressed. 

Reviewer 2 Report

In this present cohort study, Putcha and colleagues investigated verbal recognition memory performance across the amyloid- and tau-positive spectrum of Alzheimer’s Disease (AD). A total of 37 Aß+ patients were included in this study. Overall, I think this study is worthy of consideration. The topic is interesting and the manuscript is well written. I have the following minor concerns with the study.

l   The findings arising from this apparent small scale study are based on a very limited number of patients. Please demonstrate the power of your study.

l   How did the authors assess data distribution? In view of the small sample size, is it possible to use QQ plots to investigate the data distribution? Please provide QQ plots as supplementary material if applicable.

l   A paragraph demonstrating the limitations of this study would be appreciated.

l   The strength of the study should be described.

Author Response

Please see attached cover letter for responses to all reviewer comments. 
